# The sexuality of couples formed during the pandemic: An exploratory study

**Sara Filisetti**[1☉], **Carla Tortora**[1☉], **Cristina Paruta**[1], **Federica Ida Piazza**[2],
**Marta Panzeri**[2]*

1 Department of General Psychology, Padua University, Padua, Italy, 2 Department of Developmental Psychology and Socialisation, Padua University, Padua, Italy

☉ These authors contributed equally to this work.
* marta.panzeri@unipd.it

**Data Availability Statement:** Data file is available from the Research Data Unipd database at the following link: http://researchdata.cab.unipd.it/id/eprint/681 (DOI: 10.25430/researchdata.cab.unipd.it.00000681).

## Abstract

Covid-19 has been affecting people's lives on a social, economic, emotional, and sexual level. This study aims to investigate any change in how couples formed during the pandemic got to know the partner and experienced sexuality, including factors that could have influenced those changes in comparison with a pre-pandemic period. Particularly, focus groups (N = 26 women) were conducted and an online questionnaire (N = 120; 41 men and 79 women) was administered. Given the exploratory qualitative nature of the present research, no specific hypothesis was tested. Most of the sample reported an increase in sexual desire, sexual frequency, and quality of intercourse, perceiving an early development of intimacy. The results highlight the lack of stress and fear of contagion. The intense state of euphoria, typical of the initial phase of the relationship, has perhaps allowed the couples to overcome the obstacles due to the restrictions. This study underlines the role of being in love in the survival of the species, as it allows for the creation of steady relationships even in moments of danger.

## Introduction

In March 2020, the World Health Organization declared the Covid-19 epidemic a global pandemic. Given the rapid spread of the virus, governments around the world, including the Italian one, have implemented restrictive and social distancing policies with the aim of slowing down the spread of SARSCoV2. A national lockdown was established between March and May 2020 that has affected every aspect of people's daily lives. After a momentary period of easing of the restrictions during the summer, starting from October 2020, they were again implemented in Italy.

### Psychological and sexual effects of the pandemics

The increase in psychological stress and the social restrictions have also had an impact on individuals' sexual functioning [1]. Accordingly, a recent systematic review [2], showed that most of individuals have experienced a decrease in overall sexual functioning, which was related to

**Funding:** The author(s) received no specific funding for this work.

**Competing interests:** The authors have declared that no competing interests exist.

many factors including fear of transmitting COVID-19 and the increase in depression, anxiety, and stress. For example, a research conducted on the Italian population [3] showed that couples experienced negative mood associated with the discrepancy between their high sexual desire and the scarcer possibilities of actually engaging in sexual activities. On the other hand, an increase in the frequency of sexual intercourses was also reported [4], which may be explained by the fact that some couples may have kept seeing each other in person, despite social restrictions. Indeed, the presence of restrictions has damaged the sexuality of non-cohabiting couples as they have not been able to meet for weeks or even months [5]. The days spent at home, in the absence of external stimuli and the necessary privacy (either due to the constant presence of other family members or to the forced prolonged cohabitation), also influenced the relationship of cohabiting couples [2, 4, 6]. In general, most couples resorted to alternative ways of experiencing their sexuality, especially those couples who were at a distance [7]. Furthermore, the research shed light on the situation of singles who, during the pandemic, massively used social media and dating apps to find a partner [8, 9]. Moreover, generally individuals reported an increase in autoerotic sexuality and behaviours, which was reflected also in a significant increase in the consumption of pornographic material [6]. Moreover, results from the International Sexual Health and Reproductive (I-SHARE; [10]) survey suggested that people who had a steady relationship starting before COVID-19 showed an increased in sexual satisfaction. This finding is not surprising, as it could be speculated that having a steady relationship may be associated with more positive communication, less fear of transmitting COVID-19, and a reduction in mental health issues related to the pandemic and social restrictions [2, 11].

## Relationship effects of the pandemic

Interestingly, results from research on quality of relationships were mixed [2], highlighting how negative changes were predicted by the worsening of mental health conditions. Accordingly, on the other side, engaging in joint activities, communicating constructively, and being able to balance autonomy were protective factors for mental health and, in turn, for the quality of relationships [3]. Particularly, stress seems to have been a key factor, as it was found to be predictive of sexual, romantic, and individual functioning [2, 11, 12].

## Couple formation phase

Surprisingly, there is a lack of research considering newly formed relationships in comparison with longer-lasting ones. Indeed, what distinguishes established relationships from new couples is that the latter are in the initial phase of the relationship, better known as the falling in love phase. This phase is characterised by a strong sense of joy, euphoria, excitement, a constant almost obsessive need for the presence of the other [13], a state of intense energy, physiological hyperactivation (increase in levels of dopamine, noradrenaline and phenylethylamine), from a constant search for closeness to the other, but also from the fear of losing the loved one [14, 15]. In addition, a reduction in testosterone levels in men and an increase in women was noted during this phase. This hormonal variation would lead to changes in the behavioural style of both sexes: in women it would increase the occurrence of more extroverted and aggressive behaviours, on the contrary in men it would attenuate the more impetuous behavioural characteristics. It would therefore seem that the phase of falling in love allows for the temporary smoothing of gender differences, attenuating some characteristics in men and increasing the same in women [15]. In the end, the initial stages of the relationship are characterized by high levels of sexual desire, functional to the enhancement of intimacy and the formation of the couple bond [16]. Hence, the state of joy and energy, determined by falling in love, may

have allowed couples formed in the pandemic to have suffered less consequences of the pandemic, such as the fear of contagion, and anxiety from the uncertainty of the period, leading to experiencing their sexuality more positively. Additionally, to better understand how couples were affected by the social consequences of the pandemic, it is important to evaluate the factors implicated in potential changes not only in quantitative terms but also through qualitative studies, which may help explain apparently mixed results [6].

## Objectives

The aim of the present research is to fill the gap in the literature, by exploring eventual changes in the modality of formation of the relationship and in the sexuality of new couples, compared to a pre-pandemic period. In particular, we will focus on one hand on the potential changes in sexual desire, arousal, and/or frequency. On the other hand, we will focus on potential changes in the way in which the partner met, kept in touch, and in the development of the romantic relationship. In both cases, we will consider possible factors underlying the presence or absence of changes. In addition, an attempt was made to investigate the factors that led to these changes. Finally, it could be hypothesised that newly formed couples were not affected by any significant change compared to a pre-pandemic period, as this phase of the couple relationship must be preserved for the continuity of the species [17]. However, given the lack of literature on the subject, the changes were investigated through an exploratory and qualitative study, in which no specific hypothesis were tested.

## Material and methods

### Participants and procedure

The present research comprised focus groups and an online battery of questionnaires and was conducted in accordance with the ethical principles stated in the Declaration of Helsinki (2013).

For focus groups, the research group was made up of 3 people that built a conducting grid, a guide with points for reflection to be submitted to the participants (as indicated in Table 1): how they lived their sexuality during the pandemic, the perceived changes due to the pandemic, and the perceived factors influencing those changes. Focus groups were conducted from January to April 2021 according to the Morgan's suggestions [18], online through the Zoom Meetings application. Due to the impositions and restrictions of the current period, it was not possible to respect the classic circular shape, which is typical of traditional focus groups. Adult women who entered a romantic relationship from March 2020 to May 2020 were recruited through social networks, with a message containing a QR code that gave access to the link where participants could leave their details to be contacted via email. Once their willingness to participate in the research was confirmed, they were given both the informed consent and the socio-demographic questionnaire necessary to participate in the focus groups. Overall, a total of 26 women took part in the focus groups, aged between 18 and 35 (M = 23.76 and DS = 3,253), as shown in Table 2. Each session was recorded and lasted from a minimum

**Table 1. Conducting grid.**

a. Has the pandemic impacted the way your couple formed? (Modalities of getting to know the partner, distance kept, fear of contagion, first kisses)
b. How did you experience couple sexuality during the pandemic? (First physical contact with the partner, use of the mask, contraception, frequency, and types of sexual intercourse)
c. If the situation had been different, then without the pandemic and the lockdown, would you have behaved differently regarding sexuality? (Reasons for changes and reasons for non-changes)

**Table 2. Socio-demographic data of the participants.**

| | | Focus groups (N = 26) | Questionnaire (N = 120) |
|---|---|---|---|
| Provenance | Northern Italy | 19 (73%) | 108 (90%) |
| | Southern Italy | 5 (19%) | 12 (10%) |
| | Foreign | 2 (7%) | 0 (0%) |
| Employment | None | 18 (69%) | 78 (65%) |
| | Smartworking | 1 (4%) | 9 (7.5%) |
| | Traditional | 3 (11%) | 24 (20%) |
| | Mixed (smartworking/traditional) | 4 (16%) | 9 (7.5%) |
| Start of the relationship | March-May 2020 | 10 (38%) | 48 (40%) |
| | June-August 2020 | 5 (19%) | 58 (48%) |
| | September-December 2020 | 11 (42%) | 14 (12%) |
| Sexual orientation | Heterosexual | 21 (81%) | 120 (100%) |
| | Homosexual | 1 (4%) | 0 (0%) |
| | Bisexual | 4 (16%) | 0 (0%) |

of 40 minutes to a maximum of 60 minutes. Only audio tracks were kept, since they were necessary for the transcription of the focus groups. The research was approved by the Ethical Committee of Psychological Research of the University of Padua (protocol 3594). The study was reported according to the Consolidated Criteria for Reporting Qualitative Research [19].

As for the online battery of questionnaires, inclusion criteria comprised being adult and resident in Italy. Data were collected online from November 2020 to April 2021 through Qualtrics, while the recruitment took place by sharing a link accompanied by a short message introducing the research on social medias. Overall, the study lasted approximately 20 minutes. Participants took part in the research on a voluntary basis and were asked to fill a consent form, which also explained the objectives and methodology of the research, as well as that the answers were collected confidentially. Overall, 120 individuals participated in the study, aged between 19 and 46 years old (M = 23.92; SD = 3.59): 41 males aged between 19 and 31 years old (M = 24, 34 and SD = 2.36) and 79 females aged between 19 and 46 (M = 23.71 and SD = 4.08). The period in which the relationship began consists of three moments: the first lockdown (March-June 2020), June-October 2020 and the second lockdown (November 2020—April 2021). In the first lockdown we find 40% of the participants (frequency = 48), in the period between June and October 2020 48.3% with a frequency equal to 58 and, finally, in the second lockdown 11.7% of the participants (frequency = 14). This part of the study was approved by the Ethics Committee of Psychological Research of the University of Padua (protocol 3541).

## Measures

The following questionnaires were administrated.

A demographic questionnaire to investigate participants' gender, age, and sexual orientation, as well as the period in which their relationship started and the region in which they were currently living. Furthermore, it contained questions related specifically to the period of the pandemic, such as the possible contraction of the virus, the number of swabs performed, the results of the swabs, and whether the participants or their partners were working with Covid patients.

The Patient Health Questionnaire (PHQ-15 [20]; Italian version [21]), a standardized questionnaire for the screening of somatic symptoms: participants were asked to rate the level of

bother experienced during the last month about 15 specific somatic symptoms (e.g. "shortness of breath"; α = .75) on a 3-point Likert scale ranging from 0 ("not at all") to 2 ("a lot"). Thus, scores could range from 0 to 30, with somatization being considered mild with scores ≥ 5, moderate with scores ≥ 10 and severe with scores ≥ 15.

The Depression Anxiety Stress Scales-21 (DASS-21 [22]; Italian validation [23]), which consisted of 21 items divided into 3 subscale measuring symptoms of depression (e.g., "I could not experience any positive emotions"; α = .88), anxiety (e.g., "I realized that my mouth was dry"; α = .79), and stress (e.g., "I found it hard to wind down"; α = .86) on a 4-point Likert scale ranging from 0 ("never") to 3 ("almost always").

The Quality of Marriage Index (QMI [24]; although there is no Italian validation of this instrument, it is often used in the literature, e.g., [25]) to measure the overall perception of satisfaction in couple relationships. The Italian version consisted of 6 items (e.g., "My relationship with my partner makes me happy"; α = .92) rated on a 6-point Likert scale ranging from 1 ("I completely disagree") to 7 ("I completely agree").

Two factors of the Brief Index of Sexual Functioning for Women (BISF-W [26]; Italian validation [27]) and the Brief Index of Sexual function for Men (BISF-M; [28]): couple sexuality (21 items; α = .95 for women and α = .94 for men; e.g., "the frequency of sexual intercourses with a partner in the last month") and autoeroticism (6 items; α = .85 for women and α = .89 for men; e.g., "how many times in the last month did you feel the desire of masturbating on your own?"). Items were arranged in a Likert-type format, ranging from 5 to 7 points, for which participants were asked to select the most appropriate option.

## Data analysis

**Qualitative analysis.** The qualitative analysis was carried out through a thematic analysis of the verbatim transcribed focus groups [29–31]. The tool used is the data analysis software ATLAS.ti7. Through the software, the most significant issues (defined macro categories) were identified by associating precise text segments with keywords. This type of analysis made it possible to continue the transcripts under investigation to define the codes in common and add further ones. A bottom-up coding procedure was used to build the code system; it started from the text and data and based on these the open coding option was used, which allowed to associate a temporary conceptual label with the selected text segment. Through this procedure, it was possible to group text segments with similar characteristics into categories and sub-categories. The participants who named the various categories and sub-categories were counted. The use of the ATLAS.ti7 software made it possible to accurately analyse the various focus groups, highlighting issues relevant to the research.

The analyses were conducted by one of the researchers who also took care of the data collection, with the support of a third person expert in qualitative analysis who supervised the coding. Subsequently, the coding procedures were compared and discussed with a third researcher who suggested changes relating to the issues until consensus was reached. A qualitative analysis was also conducted on the open questions present at the end of the questionnaire, following the same procedure reported above.

**Statistical analysis.** Statistical analyses were performed using SPSS 25. Independent-samples t-tests were conducted on the PHQ-15, the QMI and the "couple sexuality" factor of the BISF (BISF-1) to investigate differences based on gender. Additionally, to assess differences based on the period in which the relationship started, participants who reported having started one during the second lockdown (from November 2020) were excluded from the analyses due to the small group size. Accordingly, independent-samples t-tests were conducted on the PHQ-15, the QMI, and the BISF-1 to assess differences between participants who started a

relationship during the first lockdown (T1; from March to May 2020) and those who started it during the period from June to October 2020 (T2), when restrictions were considerably loosened. Due to the small sample size only independent-samples t-tests were conducted to address whether there were differences on the QMI and the BISF-1 based on whether there was some previous acquaintance between participants and their partners.

## Results

### Qualitative analysis of focus groups

Through the focus group analysis, five macro categories were identified regarding the relational and sexual changes observed by Italian women who undertook a new relationship during the lockdown.

**Impact of the pandemic on the ways of meeting and getting to know the partner.** This theme refers to the influence that the pandemic has had on the formation of the couple and on the relational and sexual dynamics. The participants recounted their experiences regarding the ways of getting to know the partner compared to a pre-pandemic period. In particular, the experiences have changed according to the restrictions in force at the time they met the partner (first lockdown, summer period, second lockdown). Some participants told how much and if the fear of being infected or of infecting relatives affected the first physical contacts. For 14 participants (out of 26) the fear of infection has not placed limits on the relationship: after the lockdown they felt the need to experience their relationship normally even risking contracting the virus. In addition, 3 participants said they were not afraid of contracting Covid or to be able to transmit it. One participant just reported a total absence of perceived risk.

Most of the participants (24 out of 26) suffered greatly from the social limitations imposed by the government. In general, not being able to share moments of leisure with one's partner, such as going to the cinema, going for a walk, going on vacation, meeting friends or relatives, has resulted in a strong sense of lack of freedom. Additionally, 10 participants wondered how much these limitations actually impacted on making their relationship exclusive.

> *"It is more difficult to be able to integrate well with his group of friends, to get to know them, it is something that cannot be done and for example I also miss this aspect." (P13)*

During the lockdown, 5 participants reported having to use social networks or online dating apps to get to know a possible partner. In fact, four of them embarked on their relationship in this way and two of them admitted they had never used these apps before.

> *"As for how to get to know each other, yes absolutely, Tinder, because meeting in a bar or elsewhere is impossible. We met on Tinder, first date in a park, then at the supermarket and then we started going to the respective homes and that's it, we've never been to another public place, so yeah definitely different." (P11)*

**Emotional-relational consequences of distancing.** This theme collects the feelings felt during the social isolation and the distance from one's partner. Half of the participants (13) experienced the isolation and the pandemic in general with all its consequences as a useful time to reflect on the relationship that was developing and on the couple's sexuality. In fact, two participants reported that they had reached a new state of maturity and awareness of themselves and their own relational experience. In addition, there was a shift from the physical to the emotional plane that led to a strengthening of the feeling.

*"It's a paradox that I say a lot, 2020 was a bad year but it brought me several beautiful things. It's not a year that I want to throw it in the trash, on the contrary from a sentimental point of view I have found someone with whom I am really good so I'm hopeful let's say."* (P4)

When the participants were asked what were the reasons that led them to experimenting more with their partner, such as sexting or confiding sexual wishes, four of them reported developing a spirit of survival. The will of wanting to carry on the relationship, as reported by one participant, broke down walls and mental stakes that they had previously imposed on themselves.

"*One above all is the spirit of survival. Somehow we had to go on, accept the situation, internalise it and adapt to what it was. It is like this: or I die, or I evolve. Personal survival of course. Wanting to keep that relationship, I do not mean the fear that it could end but the concern of how it could go on. And then, in any case, precisely the desire that was there: at a certain point who cares about paradigms and ideas and ideologies that a person up to that moment could have had, that are of the kind: sexting you can't do it because. . . ."* (P5)

**Consequences on sexuality.** This issue refers to the influence that the pandemic has had on the sexuality of individuals and couples. The participants told their experience starting from the first approaches they had with the partner and reported differences compared to previous relationships. Overall, 12 participants experienced a noticeable acceleration and an increase in the frequency of sexual behaviours compared to the pre-pandemic period.

*"The moment we met I perceived an acceleration compared to what perhaps would have been normality, in the sense that a great expectation had been created and therefore in my opinion I noticed that engaging in sexual contact was a little quicker than what I normally would or my expectations of an ideal timing of a relationship."* (P8)

Seven participants reported that the period of abstinence given by the social distancing has allowed them to better discover their bodies through masturbation, as it was rarely practiced before the restrictions. In addition, the uncertainty and fear that new restrictions would prevent them from seeing the partner gave rise to the need for three participants to take advantage of each time available to have sex, even reluctantly. One participant spoke of an uncertainty component that causes a feeling of urgency in the relationship with the partner. Overall, 14 participants experienced sexual intercourse positively with their partner.

**Risk factors (both emotional-relational and sexuality).** The contributing reasons that caused difficulties in couple dynamics and in their sexuality have been grouped into this theme. Two participants explained how the use of the means of communication and the inability to decide when and how to see each other made getting to know the partner limiting.

"*In any case, for a long time we had to hear from each other only by text, which at first was obviously fine, then after a while it becomes limiting anyway, when enough confidence is created than you would like to carry on in person. So we felt a bit forced, after the first period, to limit ourselves to messages.*" (P6)

Those who have been able to live their relationship at home, therefore who have shared a lot of time with their partner, have experienced a sense of monotony in the long run. As

reported by six participants who would have liked to experience the relationship in other social contexts as well.

> "*I like being at home, but we are both quite dynamic people, so we like being on the move and travelling. After a while at home you're bored, so we both suffer from that, but we can't do much about it. Apart from these limitations on our lifestyle, from a relationship point of view, it has not caused any major problems.*" (P5)

**Protective factors (both emotional-relational and sexuality).** The reasons that have contributed to improving the relationship with one's partner have been grouped into this theme. As for the positive consequences of the lockdown on relational dynamics, seven participants reported that the pandemic context, that included spending a lot of time alone with the partner and managing to keep in touch even in moments of distance allowed for an open and direct discussion on their emotions, sensations, difficulties, but also on their feelings. All this had a positive effect on the strengthening of the bond that was being created and on the increase of mutual trust, in fact seven participants explained that they positively experienced an increase in intimacy with their partner given by sharing moments of daily life together, such as cooking, watching TV but also having sex at any time. Overall, 12 participants said they experienced the increase in sexual desire positively. Some have reported increased desire both during the lockdown (when they couldn't see the partner) and when they got to meet with their partner after time.

> " *The difference I see is that, when we are together, since I don't live with him, there is much more desire, because even in public with this situation certain things are done even less. Maybe you have more desire, because there is more intimacy in the private sphere.*" (P8)

Regarding the urge to want to be with one's partner despite the prohibitions and the increase in infections, 11 participants admitted to having broken the rules to spend some time together with the partner. Specifically, when only kin could see each other, three participants pretended to have been in a stable relationship.

## Qualitative analysis of open questions in the questionnaire

The qualitative analysis of the open questions of the questionnaire referred to the consequences that the pandemic had on sexuality. In fact, a single macro-category has been identified, called "Consequences on sexuality" characterised by both positive and negative aspects. Most of the participants, especially girls, felt that these consequences were caused more by the restrictions and thus by the distance from the partner.

> "*The restrictions in force with the DPCM (curfews, travel bans, closed rooms, etc.) have meant that the possibility of seeing each other or meeting new people diminished.*"

Furthermore, the pandemic period has led most of the participants to experience sexuality with stress and anxiety.

> "*Due to recurrent anxiety and stressful thoughts, I have had difficulty reaching orgasm from time to time*".

**Table 3. Results from the independent-samples t-tests.**

|  | Men (N = 41) | | Women (N = 79) | | | | |
|---|---|---|---|---|---|---|---|
|  | M. | S.D. | M | S.D. | T | df | p |
| PHQ-15 | 18.76 | 3.23 | 21.09 | 3.92 | -3.27 | 118 | .001 |
| QMI | 6.53 | 3.49 | 7.30 | 3.02 | -1.24 | 118 | .217 |
| BISF-1 | 3.43 | 1.05 | 3.43 | .90 | .02 | 70.63 | .985 |

Some subjects, particularly the female gender, reported experiencing greater arousal and an increased desire for sexual intercourse during the period of lock-down.

*"I have experienced greater mutual excitement, greater desire and satisfaction perhaps due to the fact of always being together."*

## Quantitative analyses

Results from the t-tests for independent sample showed that women obtained significantly higher scores than men ($p < .01$; $d = .65$) on the PHQ; no significant difference based on gender was found on the QMI nor on the BISF-1 (see Table 3). Frequencies of somatization levels in men and women are reported in Table 4.

From the results of the independent-samples t-tests based on the period in which the relationship started (see Table 5) it was found that individuals who started a relationship during the first lockdown had significantly higher scores on the QMI ($p < .05$; $d = 2.38$); no other significant difference could be found between the two groups.

Results from the MANOVAs computed on the DASS showed no significant difference based on the period in which the relationship started ($F_{3;102} = 1.29$; $p > .05$) nor based on gender ($F_{3;116} = 2.05$; $p > .05$). Finally, no significant difference was found based on previous acquaintance ($p > .05$) on both the BISF-1 and the QMI.

## Discussion

Through the focus group method, the changes concerning the ways of knowing the partner, the differences in the formation of the relationship and in the sexuality of the new couples with respect to a pre-pandemic period were studied. One of the objectives of the research was to explore whether the participants' experiences would change based on the restrictions in place at the time of knowledge (first lockdown, summer period, second lockdown) and on fear of

**Table 4. Frequencies of the levels of somatization based on the PHQ-15 cut offs in the male and female samples.**

|  |  | N | % | Valid % | Cumulative % |
|---|---|---|---|---|---|
| Male | Absent | 25 | 61.0 | 61.0 | 61.0 |
|  | Mild | 13 | 31.7 | 31.7 | 92.7 |
|  | Moderate | 3 | 7.3 | 7.3 | 100.0 |
|  | Severe | 0 | .0 | .0 | 100.0 |
| Female | Absent | 32 | 40.5 | 40.5 | 40.5 |
|  | Mild | 31 | 29.2 | 39.2 | 79.7 |
|  | Moderate | 13 | 16.4 | 16.4 | 96.2 |
|  | Severe | 3 | 3.80 | 3.80 | 100.0 |

**Table 5. Results from the independent-samples t-tests based on the period in which the relationship started.**

|  | T1* (N = 48) | | T2** (N = 58) | | | | |
|---|---|---|---|---|---|---|---|
|  | M. | S.D. | M | S.D. | t | df | p |
| PHQ-15 | 5.87 | 4.42 | 5.03 | 3.40 | 1.10 | 104 | .272 |
| QMI | 7.97 | 2.75 | 6.49 | 3.24 | 2.49 | 103.93 | .013 |
| BISF-1 | 3.44 | .95 | 3.51 | .86 | -.37 | 104 | .714 |

*Time 1: First lockdown (from March to May 2020)

**Time 2: From June to October 2020

contracting the virus, and/or on psychological effects of the pandemic. The results showed that isolation has failed to suppress the need to connect with each other on an emotional level and has prompted people to seek contact via the web as emerges from the literature: during the pandemic, online dating apps saw an increase in enrolments [32, 33]. More than half of the focus groups' participants reported that they were not affected by the influence of the pandemic at the time of the couple's formation as the acquaintance with the partner took place in a period in which there were no strict restrictions, e.g., between June and September. However, these participants reported that they experienced the second lockdown with greater psychological difficulty: this is in line with the existing literature which states that a second period of restriction had a negative impact on people's lives by generating uncertainty about the future [2, 34]. The social limits and an unpredictable future due to the sudden change of rules and constraints, generated in the participants a sense of doubt about the couple time ahead and consequently the fear of not being able to see each other led to a greater search for the proximity of the partner and a deeper bond. The literature shows how, in conditions of distress and stress, individuals seek the closeness of the attachment figure to receive comfort and reassurance [15, 34–37]. Quantitative results showed that there were no gender differences in the different sexual and relationship parameters, confirming that in the falling love period gender differences are levelled [15]. Together this highlight the evolutionary relevance of the phase of couple formation that seemed to overcome all possible obstacles to falling in love, deepen the relationship and create a strong sexual bonding [17].

Half of the focus groups' participants experienced isolation as a useful time to reflect on what they actually wanted from the emerging relationship and what they wanted to change in the couple's sexuality. Spending a lot of time away from ones' partner has allowed to reach a new state of maturity and self-awareness with respect to the relationship experience of the couple. In the literature this is called "silver lining" or the ability to cope with life events by finding new resources that the person thought he or she did not have [38]. Focus groups' participants who instead passed the lockdown with their partner reported how having spent more time together was a factor that allowed them to deepen the couple's relationship as not having to move and always being together had a positive impact on stress [6]. This is also confirmed by the existing literature as a good quality of relationships is able to guarantee comfort, safety and support and can generate a state of physical and emotional well-being by reducing stress [34, 39–42].

As in the case of couple formation, even concerning sexuality there were no worsening compared to a pre-pandemic period; in fact most of the focus groups' participants reported having lived sexual relations without being influenced by the fear of contagion. The literature shows that maintaining a good quality of sexual life can generate less stress and/or greater coping skills [13]. Focus groups' participants who managed to maintain sexual activity during the lockdown reported lower levels of psychological distress, developed better sexual functioning,

and improved relationship adaptation [43, 44]. Even survey participants showed no differences in anxiety, stress, or depression during the different times of the study (first lockdown, summer period, second lockdown). The lack of social contexts would have led couples to spend more time at home, even for first dates, and could have brought attention to the couple's activities to be carried out inside, including sexual activities. Additionally, those who experienced greater sexual intimacy also reported greater emotional intensity. This speed and intensity that occurred both emotionally and sexually could be explained by the fact that emotional closeness increased sexual desire towards the partner and vice versa and by the long time spent together which allowed the couple to explore more [45, 46].

During the lockdown, spending a lot of time alone with the partner and hearing frequently by phone in moments of distance allowed an open and direct discussion on one's emotions, feelings, difficulties and for this it was considered a protective factor as it had a positive response on strengthening of the bond that was being created and the increase in mutual trust. The literature shows that perceiving the partner as sensitive to one's needs is a protective factor and allows to maintain a good quality of the relationship [2, 47]. Furthermore, during the falling in love phase, moments shared with the partner facilitate the disclosure of personal information, and increase the feelings of mutual trust, until one begins to consider the partner as a safe haven and this also contributes to the growth of the relation [2, 48, 49]. The focus groups' participants were not influenced by the restrictions, in fact the desire to be with their partner, despite the prohibitions and the increase in infections, led the participants to break the rules such as pretending to have a stable relationship for a long time. This confirms the initial hypothesis as, from an evolutionary point of view, the new couple is so important for the participants that they ignore the restrictions and consequently overcome the fear of contagion.

On the contrary, a minority of focus groups' participants told how, after a few months of living together, discussions and quarrels arose. This has also been confirmed by the literature on couples formed before the pandemic: compulsory confinement and forced coexistence 24 hours a day have contributed to create new conflicts or reactivate problems already present previously, which were hidden behind the daily frenzy and social hyperactivity [1, 50–52].

Focus groups' participants stated that physical distance during the lockdown increased the desire to live the relationship in the most normal possible way, risking contracting the virus and experimenting with new practices such as sexting which has led to an improvement in sexual life compared to those who have not experienced it: those who have conducted this practice have made it maintain proximity to their partner [7]. For only a small minority of focus groups' participants, living at a distance was one of the factors that slowed down relationships and sexuality.

An acceleration in the timing of sexual intercourse and/or an increase in their frequency has therefore been reported both for those who have experienced the lockdown at a distance and for those who have experienced it together with their partner [34, 39, 40]. For ten focus groups' participants, this increase is due to more time spent together at home; for others it was a consequence of the expectations that arose during social distancing. The expectations matured during periods of distance could have led to an increase in desire and consequently in the number of sexual relations, also due to a greater need for intimacy and reassurance [2, 16, 34, 50, 53]. Furthermore, the fear that new restrictions prevented seeing one's partner may have brought out the need to exploit every available moment to have sex, as if the component of uncertainty provokes a sense of urgency in living the relationship. Coherently, female survey participants reported experiencing increased sexual arousal and desire during lockdown. These results are in line with those of the studies by Cito et al. [54], Cocci et al. [55] and Pascoal et al. [52], who report an increase in sexual activity during the lockdown, which could be due to the fact that some of the new couples have not complied with the restrictions by continuing

to see each other in person. The increase in post-quarantine sexual activity is in line with the results of Arafat et al. [4], which showed couples in the post-lockdown period conducted more intense sexual activity than before the pandemic.

The research presented several limitations. The sample size is very small, but it should be remembered that the goal of qualitative studies is not to generalize the results to the entire population, but to explore people's experience from a subjective point of view. The focus group sample is made up of women between the ages of 19 and 35 and it is necessary to consider the possibility that their young age may help explain the general good maintenance of sexual life, the violation of restrictions, positive moods and the lack of fear of contagion.

The study is retrospective, and the responses may not be entirely accurate, and may also have been influenced by the couple's status at the time of participation. Furthermore, considering specifically the methodology used, it is necessary to consider the risk of social desirability, in particular for sensitive topics such as sexuality.

Another limitation of this research concerns the methods of management. The focus groups were conducted electronically via the Zoom platform, this method had advantages as well as disadvantages: remote meetings are in fact more impersonal and make it more difficult to establish a confidential atmosphere.

## Conclusions

In light of the results, the following considerations may be drawn. First of all, implementing a mixed method allowed us to provide a wide and adequate view of the formation of couples during the pandemic period. Secondly, the results presented here showed how the need to maintain closeness despite restrictions [32], the importance of dating technology and apps [33], taking time to reflect on one's desires and expectancies in the couple, and the increase in the frequency of sexual intercourse [45] were factors that contributed to the maintenance of the couple despite the period of chaos and restriction.

Furthermore, the second lockdown was experienced with greater anguish and fatigue precisely because it was preceded by another period of isolation and uncertainty [34]. Despite this, the participants managed to find strategies that led both to strengthen their relationship and also to find new resources (eg silver lining) [38].

In conclusion, we believe that this exploratory research could have useful applications in clinical settings, as it deepens and focuses on new challenges experienced by couples formed during the pandemic as well as those who experienced problems with consequent stress during the pandemic.

Furthermore, an interesting fact that we have found is inherent to the second lockdown: in fact, this was experienced with greater anguish and fatigue precisely because behind it there was already a period of isolation and uncertainty [34]. Despite this, the participants managed to find strategies that led both to strengthen their relationship but also to find new resources (eg silverlining) [38].

In conclusion, we can believe that this exploratory research could find conditions of applicability as it allows to highlight and profile in the clinical setting new challenges experienced by couples formed during the pandemic but also for those who experienced problems with consequent stress during the pandemic.

## Author Contributions

**Conceptualization:** Marta Panzeri.

**Data curation:** Sara Filisetti.

**Formal analysis:** Carla Tortora.

**Investigation:** Cristina Paruta, Federica Ida Piazza.

**Methodology:** Marta Panzeri.

**Supervision:** Marta Panzeri.

**Writing – original draft:** Sara Filisetti, Carla Tortora, Cristina Paruta, Federica Ida Piazza.

**Writing – review & editing:** Sara Filisetti, Carla Tortora, Marta Panzeri.

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
