## [Decision Letter · Decision Letter 0]

23 Aug 2022

PONE-D-22-21780The sexuality of couples formed during the pandemic: an exploratory studyPLOS ONE

Dear Dr. Marta Panzeri,

Thank you for submitting your manuscript to PLOS ONE. After careful consideration, we feel that it has merit but does not fully meet PLOS ONE’s publication criteria as it currently stands. Therefore, we invite you to submit a revised version of the manuscript that addresses the points raised during the review process.

ACADEMIC EDITOR:Reviewers are keen on the following: the need to narrow the focus of your study and have it reflected in your introduction, use of percentages to describe the FGD sample, integration of qualitative and quantitative approaches in the Discussion section, and applicability of your findings. The specific comments by reviewers are included in this letter.

We look forward to receiving your revised manuscript.

Kind regards,

Habil Otanga, Ph.D

Academic Editor

PLOS ONE

Journal Requirements:

2. Please amend either the abstract on the online submission form (via Edit Submission) or the abstract in the manuscript so that they are identical.

Reviewers' comments:

Reviewer's Responses to Questions

**Comments to the Author**

1. Is the manuscript technically sound, and do the data support the conclusions?

Reviewer #1: Yes

Reviewer #2: Partly

2. Has the statistical analysis been performed appropriately and rigorously? 

Reviewer #1: Yes

Reviewer #2: Yes

3. Have the authors made all data underlying the findings in their manuscript fully available?

Reviewer #1: Yes

Reviewer #2: Yes

4. Is the manuscript presented in an intelligible fashion and written in standard English?

Reviewer #1: Yes

Reviewer #2: Yes

5. Review Comments to the Author

Reviewer #1: The current paper examines the effect on relationship formation and engagement in sexual behaviors in these relationships during the early phases of the COVID-19 pandemic for Italian men and women. Overall, the paper was well-written and methodologically coherent. I just have a few suggestions to strengthen and add some clarity to the paper.

The authors need to discuss the purpose of this study and how it addresses or can shed light on the various consequences of the pandemic that they describe in the literature review. Once this is identified, the authors should streamline the introduction/literature review to focus on these points. Currently, the literature review is comprehensive but too broad to understand the focus of the study. Also, the authors could use headings to identify the focus of each section in the intro/literature review.

The Authors should end with a concluding paragraph rather than the limitations.

It doesn’t look like the authors integrated their qualitative and survey findings. This should be done so that there is a context for understanding the contribution of each method and how they complement and inform one another.

Reviewer #2: I have had the privilege of reading your work. I have the following suggestions:

1. Edit your work. Check Line 4 of Introduction section - sentence beginning "On the other hand..."Word(s) missing; In Participants and Procedure section check Line 12 on p11 - word(s) missing after "given informed consent..."; In the Statistical Analysis section (p16), the last sentence provides a qualitative justification for a quantitative decision (use of t-tests). Ensure clarity.

2. Results: I have a problem with your use of percentages to describe a qualitative sample. It is difficult for the reader to conceptualize 19.2% or 92.3% of women (out of 26). For clarity, indicate frequency, and then show how you arrived at the frequency. For instance, were you counting participants who presented similar views?

3. Discussion: Be clear as to which sub-sample is being referred to in this section (whether FGD or questionnaires). The reader is confused when you say "half of the participants" or "minority of participants" without specifying whether they are FGD participants of those who responded to the quantitative tools.

Finally, let the reader have an idea as to the practicality/applicability of the findings.

6. PLOS authors have the option to publish the peer review history of their article (what does this mean?). If published, this will include your full peer review and any attached files.

Reviewer #1: No

Reviewer #2: No

---

## [Author Response · Author response to Decision Letter 0]

7 Sep 2022

We thank reviewer #1 for his/her appreciation and precious advice.

According to his/her suggestions we divide the Introduction section into subsections with headings, creating an “Objective” section to better explain the purpose of our study, adding this sentence:

“In particular, we will focus on one hand on the potential changes in sexual desire, arousal, and/or frequency. On the other hand, we will focus on potential changes in the way in which the partner met, kept in touch, and the development of a romantic relationship. In both cases, we will consider possible factors underlying the presence or absence of changes.”

We also conclude the paper with a “Conclusion” section:

“In light of the results, the following considerations may be drawn. First of all, implementing a mixed method allowed us to provide a wide and adequate view of the formation of couples during the pandemic period. Secondly, the results presented here showed how the need to maintain closeness despite restrictions (32), the importance of dating technology and apps (33), taking time to reflect on one’s desires and expectancies in the couple, and the increase in the frequency of sexual intercourse (46) were factors that contributed to the maintenance of the couple despite the period of chaos and restriction.

Furthermore, the second lockdown was experienced with greater anguish and fatigue precisely because it was preceded by another period of isolation and uncertainty (34). Despite this, the participants managed to find strategies that led both to strengthen their relationship and also to find new resources (eg silver lining) (39).

In conclusion, we believe that this exploratory research could have useful applications in clinical settings, as it deepens and focuses on new challenges experienced by couples formed during the pandemic as well as those who experienced problems with consequent stress during the pandemic.”

We integrated qualitative and quantitative results in more points:

“Quantitative results showed that there were no gender differences in the different sexual and relationship parameters, confirming that in the falling love period gender differences are leveled (15). Together this highlights the evolutionary relevance of the phase of couple formation that seemed to overcome all possible obstacles to falling in love, deepen the relationship and create a strong sexual bonding (17)”

“[Focus groups’ participants who managed to maintain sexual activity during the lockdown reported lower levels of psychological distress, developed better sexual functioning, and improved relationship adaptation (44,45)]. Even survey participants showed no differences in anxiety, stress, or depression during the different times of the study (first lockdown, summer period, second lockdown).”

“[Furthermore, the fear that new restrictions prevented seeing one's partner may have brought out the need to exploit every available moment to have sex, as if the component of uncertainty provokes a sense of urgency in living the relationship.] Coherently, female survey participants reported experiencing increased sexual arousal and desire during the lockdown.”

We thank reviewer #2 for his/her appreciation of our work and his/her suggestions:

We answer point to point to him/her:

1. Edit your work. Check Line 4 of Introduction section - sentence beginning "On the other hand..."Word(s) missing;

We changed “increased” into “increase”: “an increased in the frequency of sexual intercourses was also reported”

 In Participants and Procedure section check Line 12 on p11 - word(s) missing after "given informed consent..."

We added “both”: “they were given both the informed consent and the socio-demographic questionnaire”

In the Statistical Analysis section (p16), the last sentence provides a qualitative justification for a quantitative decision (use of t-tests). Ensure clarity.

We changed it into: “Due to the small sample size only”

2. Results: I have a problem with your use of percentages to describe a qualitative sample. It is difficult for the reader to conceptualize 19.2% or 92.3% of women (out of 26). For clarity, indicate frequency, and then show how you arrived at the frequency.

We changed all percentages into frequencies. 

For instance, were you counting participants who presented similar views? 

Thank you for rise this point, we missed to explain that. Now we added in the Method section: “The participants who named the various categories and sub-categories were counted.”

3. Discussion: Be clear as to which sub-sample is being referred to in this section (whether FGD or questionnaires). The reader is confused when you say "half of the participants" or "minority of participants" without specifying whether they are FGD participants of those who responded to the quantitative tools. 

We now added focus groups’ participants where needed. 

Finally, let the reader have an idea as to the practicality/applicability of the findings.

We add a conclusion section in which we explain the applicability of the findings:

“In light of the results, the following considerations may be drawn. First of all, implementing a mixed method allowed us to provide a wide and adequate view of the formation of couples during the pandemic period. Secondly, the results presented here showed how the need to maintain closeness despite restrictions (32), the importance of dating technology and apps (33), taking time to reflect on one’s desires and expectancies in the couple, and the increase in the frequency of sexual intercourse (46) were factors that contributed to the maintenance of the couple despite the period of chaos and restriction.

Furthermore, the second lockdown was experienced with greater anguish and fatigue precisely because it was preceded by another period of isolation and uncertainty (34). Despite this, the participants managed to find strategies that led both to strengthen their relationship and also to find new resources (eg silver lining) (39).

In conclusion, we believe that this exploratory research could have useful applications in clinical settings, as it deepens and focuses on new challenges experienced by couples formed during the pandemic as well as those who experienced problems with consequent stress during the pandemic.”

---

## [Editor Report · Decision Letter 1]

12 Sep 2022

The sexuality of couples formed during the pandemic: an exploratory study

PONE-D-22-21780R1

Dear Dr. Panzeri,

We’re pleased to inform you that your manuscript has been judged scientifically suitable for publication and will be formally accepted for publication once it meets all outstanding technical requirements.

Kind regards,

Habil Otanga, Ph.D

Academic Editor

PLOS ONE
---

## [Editor Report · Acceptance letter]

20 Sep 2022

PONE-D-22-21780R1 

The sexuality of couples formed during the pandemic: an exploratory study 

Dear Dr. Panzeri:

I'm pleased to inform you that your manuscript has been deemed suitable for publication in PLOS ONE. Congratulations! Your manuscript is now with our production department. 

Kind regards, 

on behalf of

Dr. Habil Otanga 

Academic Editor

PLOS ONE